# Variational Hashing-based Collaborative Filtering with Self-Masking

## Abstract

Hashing-based collaborative filtering learns binary vector representations (hash codes) of users and items, such that recommendations can be computed very efficiently using the Hamming distance, which is simply the sum of differing bits between two hash codes. A problem with hashing-based collaborative filtering using the Hamming distance, is that each bit is equally weighted in the distance computation, but in practice some bits might encode more important properties than other bits, where the importance depends on the user. To this end, we propose an end-to-end trainable variational hashing-based collaborative filtering approach that uses the novel concept of *self-masking*: the user hash code acts as a mask on the items (using the Boolean AND operation), such that it learns to encode which bits are important to the user, rather than the user's preference towards the underlying item property that the bits represent. This allows a binary *user-level* importance weighting of each item without the need to store additional weights for each user. We experimentally evaluate our approach against state-of-the-art baselines on 4 datasets, and obtain significant gains of up to 12% in NDCG. We also make available an efficient implementation of *self-masking*, which experimentally yields <4% runtime overhead compared to the standard Hamming distance.

## 1 Introduction

Collaborative filtering (Herlocker et al., 1999) is an integral part of personalized recommender systems and works by modelling user preference on past item interactions to predict new items the user may like (Sarwar et al., 2001). Early work is based on matrix factorization approaches (Koren et al., 2009) that learn a mapping to a shared $m$-dimensional real-valued space between users and items, such that user-item similarity can be estimated by the inner product. The purpose of *hashing-based* collaborative filtering (Liu et al., 2014) is the same as traditional collaborative filtering, but allows for fast similarity searches to massively increase efficiency (e.g., realtime brute-force search in a billion items (Shan et al., 2018)). This is done by learning semantic hash functions that map users and items into binary vector representations (*hash codes*) and then using the Hamming distance (the sum of differing bits between two hash codes) to compute user-item similarity. This leads to both large storage reduction (floating point versus binary representations) and massively faster computation through the use of the Hamming distance.

One problem with hashing-based collaborative filtering is that each bit is weighted equally when computing the Hamming distance. This is a problem because the importance of each bit in an item hash code might differ between users. The only step towards addressing this problem has been to associate a weight with $k$-bit blocks of each hash code (Liu et al., 2019). However, smaller values of $k$ lead to increased storage cost, but also significantly slower computation due to the need of computing multiple weighted Hamming distances. To solve this problem, without using any additional storage and only a marginal increase in computation time, we present **Va**riational **H**ashing-based collaborative filtering with **S**elf-**M**asking (VaHSM-CF). VaHSM-CF is our novel variational deep learning approach for hashing-based collaborative filtering that learns hash codes optimized for *self-masking*. Self-masking is a novel technique that we propose in this paper for *user-level* bit-weighting on all items. Self-masking modifies item hash codes by applying an AND operation between an item and user hash code, before computing the standard Hamming distance between the user and self-masked item hash codes. Hash codes optimized with self-masking represent which bit-dimensions encode properties that are *important* for the user (rather than a bitwise -1/1 *preference* towards each

property). In practice, when ranking a set of items for a specific user, self-masking ensures that only bit differences on bit-dimensions that are equal to 1 for the user hash code are considered, while ignoring the ones with a -1 value, thus providing a *user-level* bitwise binary weigthing. Since self-masking is applied while having the user and item hash codes in the lowest levels of memory (i.e., register), it only leads to a very marginal efficiency decrease.

We **contribute** (i) a new variational hashing-based collaborative filtering approach, which is optimized for (ii) a novel self-masking technique, that outperforms state-of-the-art baselines by up to 12% in NDCG across 4 different datasets, while experimentally yielding less than 4% runtime overhead compared to the standard Hamming distance. We publicly release the code for our model, as well as an efficient implementation of the Hamming distance with self-masking[1].

## 2  RELATED WORK

We focus on collaborative filtering with *explicit* feedback, which assumes that users and items are related via a user-specified rating: the task is to rank a pool of pre-selected items. This is different from *implicit* feedback, where the task is to estimate the pool of items that are of interest to the user. Matrix factorization is one of the most successful collaborative filtering methods (Koren et al., 2009), but to reduce storage requirements and speed up computation, hashing-based collaborative filtering has been researched. For hashing-based methods the users and items are represented as binary hash codes (as opposed to real-valued vectors), such that the highly efficient Hamming distance (as opposed to the inner product) can be used for computing user-item similarities.

**Two-stage approaches.** Early hashing-based collaborative filtering methods include two stages: First, real-valued user and item vectors are learned, and then the real-valued vectors are transformed into binary hash codes. Zhang et al. (2014) employ matrix factorization initially, followed by a binary quantization of rounding the real-valued vectors, while ensuring that the hash code is preference preserving of the observed user-item ratings using their proposed Constant Feature Norm constraint. Zhou & Zha (2012) and Liu et al. (2014) both explore binary quantization strategies based on orthogonal rotations of the real-valued vectors, which share similarities with Spectral Clustering (Yu & Shi, 2003). However, the two-stage approaches often suffer from large quantization errors (Zhang et al., 2016; Liu et al., 2019), because the hash codes are not learned directly, but rather based on different quantization procedures.

**Learned hashing approaches.** Zhang et al. (2016) propose Discrete Collaborative Filtering (DCF), which is a binary matrix factorization approach that directly learns the hash codes using relaxed integer optimization, while enforcing bit balancing and decorrelation constraints. Extensions of DCF have focused on incorporating side-information (e.g., reviews associated with a rating) (Lian et al., 2017; Liu et al., 2018; Zhang et al., 2019) and have been redesigned for *implicit* feedback signals (Zhang et al., 2017). More recent work addresses the problem that hashing-based collaborative filtering methods have reduced representational power compared to real-valued vectors, but increasing the hash code dimensionality to match the amount of bits used in the real-valued case hurts model generalization (Liu et al., 2019). To address this, Liu et al. (2019) propose Compositional Coding for Collaborative Filtering (CCCF), which is broadly similar to learning compositional codes for (word) embedding compression (Chen et al., 2018; Shu & Nakayama, 2018). CCCF is a hybrid approach that combines hash codes and real-valued weights: each hash code is split into $k$ blocks of $r$ bits each, and each block is associated with a real-valued scalar indicating the *weight* of the block. The distance between two CCCF hash codes is then computed as a weighted sum of the Hamming distances of the individual blocks, where each weight is the product of each block's weight. The problem with this approach is that each block requires an individual Hamming distance computation, as well as floating point multiplications of the block weights. In fact, the CCCF block construction no longer allows for highly efficient Boolean operations because the distance computation is weighted by each block's weight. Another problem with CCCF is that it drastically increases storage requirements by needing to store the real-valued weights for all blocks in a hash code.

In contrast to CCCF, our proposed variational hashing-based collaborative filtering with self-masking solves the same problems, as it effectively allows to disable unimportant bits – corresponding to a 1-bit block size with 0/1 weights – without needing to store any additional weights

---

[1]The code is available at `anonymized-for-submission`

or vectors. Additionally, after having applied the self-masking, user-item similarity can still be computed using only a single Hamming distance on the two hash codes.

## 3 HASHING-BASED COLLABORATIVE FILTERING

Hashing-based Collaborative filtering aims to learn binary user and item representations (called *hash codes*), such that the distance between the representations indicates how well user $u$ likes item $i$. In practice, the Hamming distance is used due to its fast hardware-level implementation. Formally, we learn $z_u \in \{-1, 1\}^m$ and $z_i \in \{-1, 1\}^m$, where $m$ is the number of bits in the hash code, which is typically chosen to fit into a machine word. The preference of user $u$ for item $i$ is specified by the rating $R_{u,i} \in \{1, 2, 3, ..., K\}$, where $K$ is the maximum rating, such that the Hamming distance between $z_u$ and $z_i$ is low when $R_{u,i}$ is high. Computing the Hamming distance computation is extremely efficient due to fast hardware-level implementation of the Boolean operations, as

$$\text{Hamming}(z_u, z_i) = \text{SUM}(z_u \text{ XOR } z_i) \tag{1}$$

where SUM is computed fast on hardware using the *popcnt* instruction. Given a user and set of items, the integer-valued Hamming distances can be linear-time sorted using e.g. radix sort (because Hamming distances are bounded in $[0, m]$) in ascending order to create a ranked list based on user preference (Shan et al., 2018).

### 3.1 SELF-MASKING

The Hamming distance assigns equal weights to all bits, but in reality bit importance might differ among users. For example, if we consider each bit an encoding of a specific property of an item (e.g., a movie being a thriller), then the weight of each property would be dependent on each user's preference. However, since the hash codes are binary, it is not possible to encode such preference weights without using more storage and computation time due to no longer operating on binary values. In fact, no existing method even allows disabling specific bits for certain users (corresponding to the case of 0 preference weight). We next present a solution for the latter problem, which encodes the importance of each bit directly into the user hash code, and therefore does not require any additional storage. We define the Hamming distance with *self-masking*:

$$\text{Hamming}_{\text{self-mask}}(z_u, z_i) = \text{SUM}(z_u \text{ XOR } \underbrace{(z_i \text{ AND } z_u)}_{\text{self-masking}}) \tag{2}$$

We first apply an AND operation between the user and item hash codes, and then compute the Hamming distance between that and the user hash code. This fundamentally changes the purpose of the user hash code: instead of encoding a positive or negative *preference* for a property, it encodes which properties are *important* to the user (-1's from the user hash code are copied to the item due to the AND operation). This allows the model to disable unimportant bits on a *user-level*, meaning that we enable the model to produce user-specific item representations while still only storing a single hash code for each user and item respectively. Self-masking requires an additional Boolean operation, but since this is applied once the hash codes are already placed in the lowest levels of memory (i.e., register), it only leads to a marginal decrease in efficiency (see Section 4.6 for an empirical analysis).

### 3.2 VARIATIONAL HASHING-BASED COLLABORATIVE FILTERING

To derive a variational setup for hashing-based collaborative filtering, we define the likelihood of a user, $u$, and the likelihood of an item $i$, as the product over the likelihoods of the observed user specified ratings:

$$p(u) = \prod_{i \in I} p_\theta(R_{u,i}) \tag{3}$$

$$p(i) = \prod_{u \in U} p_\theta(R_{u,i}) \tag{4}$$

where $\theta$ are the parameters of the (neural network) model. This formulation enforces a dual symmetric effect of users being defined by all their rated items, and items being defined by the ratings

provided by all the users. To maximize the likelihood of all observed items and users, we need to maximize the likelihood of the observed ratings $p_\theta(R_{u,i})$. Note that instead of maximizing the raw likelihood, we consider the log likelihood to derive the objective below. We assume that the likelihood of a rating, $p_\theta(R_{u,i})$, is conditioned on two latent vectors, a user hash code $z_u$, and an item hash code $z_i$. To obtain the hash codes of the user and item, we assume that $z_u$ and $z_i$ each are sampled by repeating $m$ Bernoulli trials, which have equal probability of sampling -1 and 1. This gives us the following log likelihood, which we wish to maximize:

$$\log p_\theta(R_{u,i}) = \log \sum_{\substack{z_i \in \{-1,1\}^m \\ z_u \in \{-1,1\}^m}} p_\theta(R_{u,i}|z_u, z_i)p(z_i)p(z_u) \tag{5}$$

The latent vectors $z_u$ and $z_i$ are a user and item hash code, and should therefore be conditioned on the user and item respectively. To do this we first multiply and divide with the approximate posterior distributions $q_\phi(z_i|i)$, $q_\psi(z_u|u)$:

$$\log p_\theta(R_{u,i}) = \log \sum_{\substack{z_i \in \{-1,1\}^m \\ z_u \in \{-1,1\}^m}} p_\theta(R_{u,i}|z_u, z_i)p(z_i)p(z_u)\frac{q_\phi(z_i|i)}{q_\phi(z_i|i)}\frac{q_\psi(z_u|u)}{q_\psi(z_u|u)} \tag{6}$$

where $\psi$ and $\phi$ are the parameters of the approximate posteriors. We can now rewrite to an expectation and apply Jensen's inequality to obtain a lower bound on the log likelihood:

$$\log p_\theta(R_{u,i}) \geq \mathbb{E}_{q_\phi(z_i|i),q_\psi(z_u|u)} \log \left[ p_\theta(R_{u,i}|z_u, z_i)\frac{p(z_i)}{q_\phi(z_i|i)}\frac{p(z_u)}{q_\psi(z_u|u)} \right]$$
$$= \mathbb{E}_{q_\phi(z_i|i),q_\psi(z_u|u)} \Big[ \log \left[ p_\theta(R_{u,i}|z_u, z_i) \right] + \log p(z_i) - \log q_\phi(z_i|i)$$
$$+ \log p(z_u) - \log q_\psi(z_u|u) \Big] \tag{7}$$

Since $z_i$ and $z_u$ will be sampled independently, then $q_\phi(z_i|i)$ and $q_\psi(z_u|u)$ are independent and we can rewrite to the variational lower bound:

$$\log p_\theta(R_{u,i}) \geq \mathbb{E}_{q_\phi(z_i|i),q_\psi(z_u|u)} \Big[ \log[p_\theta(R_{u,i}|z_u, z_i)] \Big]$$
$$- \mathrm{KL}(q_\phi(z_i|i), p(z_i)) - \mathrm{KL}(q_\psi(z_u|u), p(z_u)) \tag{8}$$

where $\mathrm{KL}(\cdot, \cdot)$ is the Kullback-Leibler divergence. Thus, to maximize the expected log likelihood of the observed rating, we need to maximize the conditional log likelihood of the rating, while minimising the KL divergence between the approximate posterior and prior distribution of the two latent vectors. Maximizing the expected conditional log likelihood can be considered as a reconstruction term of the model, while the KL divergence can be considered as a regularizer.

Next we present the computation of the approximate posterior distributions $q_\phi^i(z_i|i)$ and $q_\psi^u(z_u|u)$ (Section 3.3) and the conditional log likelihood of the rating $p_\theta(R_{u,i}|z_u, z_i)$ (Section 3.4).

### 3.3 Computing the approximate posterior distributions

The approximate posterior distributions can be seen as two encoder functions modelled through a neural network by considering the functions as embedding layers. Each encoder function maps either a user or an item to a hash code. Next, we focus on the derivation of the encoder function for the user, as they are both computed in the same way. The probability of the $j$'th bit is given by:

$$q_\phi^{(j)}(z_u|u) = \sigma(E_u^{(j)}) \tag{9}$$

where $E_u^{(j)}$ is the $j$'th entry in a learned real-valued embedding $E$ for user $u$, and $\sigma$ is the sigmoid function. The $j$'th bit is then given by:

$$z_u^{(j)} = \lceil \sigma(E_u^{(j)}) - \mu^{(j)} \rceil \tag{10}$$

where $\mu^{(j)}$ is either chosen stochastically by sampling $\mu^{(j)}$ from a uniform distribution in the interval [0,1], or chosen deterministically to be 0.5, which can be used for evaluation to obtain fixed hash codes. Note that $\mu$ is sampled for each bit. As the sampling is non-differentiable, a straight-through estimator (Bengio et al., 2013) is used for backpropagation.

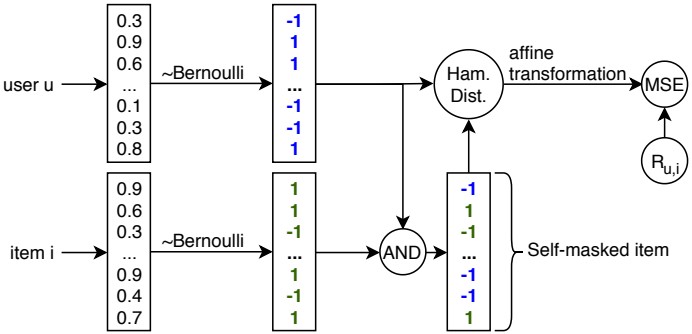

Figure 1: Model overview: The $m$-dimensional user and item sampling probabilities are learned and then sampled by repeating $m$ Bernoulli trials. The AND operation denotes the self-masking. The model is optimized using MSE between an affine transformation of the Hamming distance and the observed rating.

### 3.4 COMPUTING THE CONDITIONAL LOG LIKELIHOOD

The conditional log likelihood can be considered a reconstruction of the rating, given the user and item hash codes. We model the observed ratings as a ground truth rating with additive standard normally distributed noise, which is then discretized to the observed categorical rating. The conditional log likelihood can then be computed as:

$$p_\theta(R_{u,i}|z_u, z_i) = \mathcal{N}(R_{u,i} - f(z_u, z_i), \sigma^2) \qquad (11)$$

where $f(z_u, z_i)$ is a function that reconstructs the rating given the user and item hash codes. Maximising the log likelihood, $\log p_\theta(R_{u,i}|z_u, z_i)$, corresponds to minimising the mean squared error (MSE) between $R_{u,i}$ and $f(z_u, z_i)$, which is done for training the model. Existing work on hashing-based collaborative filtering (Liu et al., 2019; Zhang et al., 2016) also employs a MSE objective, and thus implicitly make the same normal distribution assumption as done in this work.

We define the reconstruction function to be the self-masking Hamming distance from Eq. 2:

$$f(z_u, z_i) = g(\text{Hamming}_{\text{self-mask}}(z_u, z_i)) \qquad (12)$$

where $g$ is a fixed affine transformation that maps the interval of the Hamming distance to the interval of the ratings, such that the minimum and maximum of the Hamming distance correspond to the minimum and maximum of the ratings. The model is now fully differentiable and can be trained end-to-end using backpropagation, such that the network is able to optimize the hash codes directly for self-masking. A depiction of the model is provided in Figure 1. It should be noted that while variational autoencoders are generative models, we do not explicitly utilize this in our model, and are primarily concerned with the reconstruction of the observed ratings.

## 4 EXPERIMENTAL EVALUATION

### 4.1 DATASETS AND EVALUATION METRICS

We evaluate on 4 publicly available datasets commonly used in prior work (Zhang et al., 2016; Liu et al., 2019; Zhang et al., 2017; Liu et al., 2018; Lian et al., 2017) and summarized in Table 1. Specifically, we use: two movie rating datasets, Movielens 1M[2] (ML-1M) and Movielens 10M[3] (ML-10M); a Yelp dataset with ratings of e.g., restaurant and shopping malls[4]; and a book rating dataset from Amazon[5](He & McAuley, 2016). Similarly to Rendle et al. (2009), we filter the data such that all users and items have at least 10 ratings. Following Zhang et al. (2016), for each user

---

[2]https://grouplens.org/datasets/movielens/1m/

[3]https://grouplens.org/datasets/movielens/10m/

[4]https://www.yelp.com/dataset/challenge

[5]http://jmcauley.ucsd.edu/data/amazon/

| Dataset | #Ratings | #Items | #Users | Density | Range of ratings |
|---------|----------|--------|--------|---------|------------------|
| ML-1M | 998,539 | 3,260 | 6,040 | 5.07% | 1-5 |
| ML-10M | 9,995,471 | 9,708 | 69,878 | 1.47% | 1-10 |
| Yelp | 602,517 | 14,873 | 22,087 | 0.18% | 1-5 |
| Amazon | 4,701,968 | 128,939 | 158,650 | 0.02% | 1-5 |

Table 1: Dataset statistics. Density: % ratings relative to the total amount of user-item combinations.

50% of the ratings are used for testing, 42.5% are used for training, while the last 7.5% are used for validation.

We evaluate our method, VaHSM-CF, and all baselines (see Section 4.2) using Normalised Discounted Cumulative Gain (NDCG) (Järvelin & Kekäläinen, 2000), which is often used to evaluate recommender systems with non-binary ratings (or relevance values). We use the average NDCG at cutoffs $\{2, 6, 10\}$ over all users and report the average for each cutoff value.

## 4.2 BASELINES

We use as baselines the state-of-the-art methods for hashing-based collaborative filtering (see Section 2), standard matrix factorisation, and two different strategies for binarising the output of the matrix factorisation[6]. We include the standard matrix factorization as a reference to a traditional real-valued collaborative filtering baseline, in order to highlight the performance gap between the hashing-based approaches and a real-valued approach.

**DCF**[7] (Zhang et al., 2016) learns user and item hash codes through a binary matrix factorization solved as a relaxed integer problem.

**CCCF**[8] (Liu et al., 2019) learns hash codes consisting of $k$ blocks, where each block has $r$ bits. A floating point weight is associated with each block for computing user-item similarities as a weighted sum of block-level Hamming distances. In the original paper, the floating point weights are not counted towards the amount of bits used, thus leading to an unfair advantage. For a fair comparison, we count each floating point weight as 16 bits in the following experimental comparison.

**MF**[9] (Koren et al., 2009) is the classical matrix factorization based collaborative filtering method, where latent real-valued vectors are learned for users and items. We set the latent dimension to be the same as the number of bits used in the hashing-based approaches.

**MF**$_{\mathbf{mean}}$ and **MF**$_{\mathbf{median}}$ are based on MF, but use either each dimension's mean or median for doing the binary quantization to bits (Zhang et al., 2010). We include these to highlight the large quantization loss occurring when the hash codes are not learned directly.

**VaH-CF** is our proposed method without self-masking. We use it to show the effect of a neural variational hashing-based collaborative approach without self-masking.

## 4.3 TUNING

We train both VaHSM-CF and VaH-CF using the Adam optimizer (Kingma & Ba, 2014), and tune the learning rate from the set $\{0.005, 0.001, 0.0005\}$, where 0.001 is always chosen across all data sets. The batch size is chosen from the set $\{100, 200, 400, 800\}$, where 400 was always chosen. To reduce over-fitting, Gaussian noise is added to the ratings during training, as noise injection has been found beneficial in multiple domains for variational neural models (Sohn et al., 2015). For the noise injection, we initially set the variance of the Gaussian to 1 and reduce by a factor of $1 - 10^{-4}$ every iteration during training. Our model is implemented using the Tensorflow Python library (Abadi et al., 2016), and all experiments are run on Titan X GPUs.

All hyper parameters for the baselines are tuned using the same set of possible values as in the original papers. For the CCCF baseline, we consider block sizes of $\{8, 16, 32, 64\}$ and each floating

---

[6]All hyperparameters are tuned on the validation data as described in the original papers.
[7]https://github.com/hanwangzhang/Discrete-Collaborative-Filtering
[8]https://github.com/3140102441/CCCF
[9]Provided as a baseline in the CCCF repository https://github.com/3140102441/CCCF

| 32 bits | ML-1M | | | ML-10M | | | Yelp | | | Amazon | | |
|---|---|---|---|---|---|---|---|---|---|---|---|---|
| | NDCG@2 | NDCG@6 | NDCG@10 | NDCG@2 | NDCG@6 | NDCG@10 | NDCG@2 | NDCG@6 | NDCG@10 | NDCG@2 | NDCG@6 | NDCG@10 |
| $MF_{mean}$ | 0.5655 | 0.5806 | 0.6042 | 0.3788 | 0.4190 | 0.4665 | 0.6003 | 0.7192 | 0.7708 | 0.7231 | 0.8011 | 0.8376 |
| $MF_{median}$ | 0.5659 | 0.5792 | 0.6021 | 0.3793 | 0.4190 | 0.4665 | 0.6035 | 0.7194 | 0.7717 | 0.7195 | 0.7977 | 0.8348 |
| DCF | 0.6730 | 0.6875 | 0.7088 | 0.5275 | 0.5618 | 0.6009 | 0.6588 | 0.7642 | 0.8080 | 0.7737 | 0.8382 | 0.8681 |
| CCCF | 0.6507 | 0.6768 | 0.7003 | 0.5227 | 0.5583 | 0.5982 | 0.6417 | 0.7506 | 0.7978 | - | - | - |
| VaH-CF | 0.6755 | 0.6916 | 0.7137 | 0.5382* | 0.5745* | 0.6137* | 0.6668* | 0.7694* | 0.8124* | 0.7795* | 0.8419* | 0.8712* |
| VaHSM-CF | **0.7362*** | **0.7304*** | **0.7405*** | **0.5555*** | **0.5892*** | **0.6249*** | **0.7424*** | **0.8176*** | **0.8517*** | **0.8081*** | **0.8618*** | **0.8874*** |
| MF (real-valued) | 0.7373 | 0.7367 | 0.7502 | 0.5915 | 0.6107 | 0.6428 | 0.7553 | 0.8238 | 0.8564 | 0.8183 | 0.8686 | 0.8929 |

| 64 bits | ML-1M | | | ML-10M | | | Yelp | | | Amazon | | |
|---|---|---|---|---|---|---|---|---|---|---|---|---|
| | NDCG@2 | NDCG@6 | NDCG@10 | NDCG@2 | NDCG@6 | NDCG@10 | NDCG@2 | NDCG@6 | NDCG@10 | NDCG@2 | NDCG@6 | NDCG@10 |
| $MF_{mean}$ | 0.5714 | 0.5826 | 0.6069 | 0.3825 | 0.4211 | 0.4675 | 0.6050 | 0.7193 | 0.7716 | 0.7238 | 0.8007 | 0.8374 |
| $MF_{median}$ | 0.5684 | 0.5805 | 0.6049 | 0.3835 | 0.4217 | 0.4679 | 0.6067 | 0.7230 | 0.7752 | 0.7205 | 0.7983 | 0.8354 |
| DCF | 0.6922 | 0.7043 | 0.7242 | 0.5528 | 0.5824 | 0.6193 | 0.6708 | 0.7725 | 0.8148 | 0.7788 | 0.8418 | 0.8710 |
| CCCF | 0.6862 | 0.7008 | 0.7220 | 0.5390 | 0.5735 | 0.6127 | 0.6528 | 0.7584 | 0.8039 | - | - | - |
| VaH-CF | 0.7014* | 0.7152* | 0.7340* | 0.5621* | 0.5960* | 0.6332* | 0.6769* | 0.7759* | 0.8178* | 0.7840* | 0.8446* | 0.8733* |
| VaHSM-CF | **0.7204*** | **0.7277*** | **0.7441*** | **0.5815*** | **0.6072*** | **0.6399*** | **0.7511*** | **0.8224*** | **0.8553*** | **0.8162*** | **0.8663*** | **0.8911*** |
| MF (real-valued) | 0.7440 | 0.7438 | 0.7552 | 0.5905 | 0.6101 | 0.6423 | 0.7541 | 0.8248 | 0.8573 | 0.8235 | 0.8711 | 0.8948 |

Table 2: NDCG@K of our method (VaHSM-CF) against baselines using hash codes of length 32 and 64 bits. The missing results for CCCF on Amazon are due to the model requiring more than 128 GB of RAM. Statistically significant improvements using a paired two tailed t-test at the 0.05 level, compared to the best existing hashing-based baseline (DCF), are indicated by *.

point weight counts for 16 bits. We try all possible combinations that fit within the bit budget, and if a single block is chosen, then the weight is not included in the bit calculation.

## 4.4 RESULTS

The results are shown in Table 2, for hash code lengths of 32 and 64 bits, as these correspond to common machine word sizes. The highest NDCG per column is shown in **bold** (among hashing-based baselines), and results statistically significantly better than the best hashing-based baseline (DCF), using a paired two tailed t-test at the 0.05 level, are indicated by an asterisk *. The Amazon results for CCCF are not included, as the released implementation requires excessive amounts of RAM ($>$128GB) on this dataset due to the large amount of items and users.

Our proposed VaHSM-CF significantly outperforms all hashing-based baselines across all datasets by up to 12%. Our VaH-CF without self-masking is second best (up to 2% better than the hashing-based baselines), although on ML-1M the VaH-CF results are not statistically significantly better than DCF. This highlights the benefit of modelling hashing-based collaborative filtering with a variational deep learning based framework, which is notably different from existing hashing-based methods based on dicrete matrix factorization solved as relaxed integer problems. Most importantly however, it shows the significant improvement self-masking brings to hashing-based collaborative filtering. We observe that the top 3 baselines (including our VaH-CF) generally obtain similar scores, which highlights the difficulty of improving performance without changing how the hash codes are used (as done by self-masking).

The real-valued MF baseline outperforms all the hashing-based approaches, which is to be expected as the representational power of of 32/64 floating point numbers are notably higher than 32/64 bits. However, our VaHSM-CF bridges a large part of the gap between existing hashing-based methods and MF, such that the NDCG difference in most cases is below 0.01. Additionally, the large performance decrease by both the mean or median rounding shows the large quantization error obtained if the hash codes are not learned directly, as done by our and the existing hashing-based approaches.

## 4.5 MODEL ANALYSIS

**How self-masking influences the convergence rate of the model.** Figure 2a shows the convergence rate for the ML-1M dataset with and without self-masking. We see that training with self-masking significantly improves the convergence rate compared to the model without self-masking. Since the time for a single epoch is approximately the same with and without self-masking, we conclude that self-masking not only improves NDCG, but also reduces training time by a very large margin.

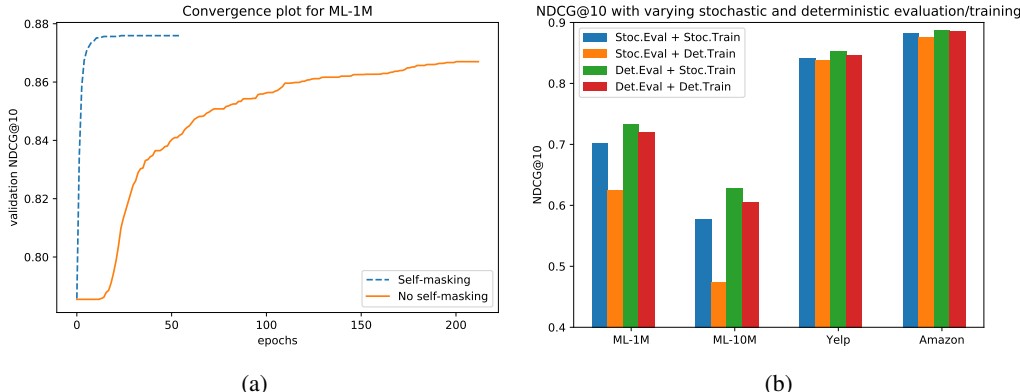

(a)                                          (b)

Figure 2: 2a shows convergence using the validation NDCG@10 on ML-1M, where self-masking significantly speeds up convergence. We observe the same trend on the other datasets (see Appendix A.1). 2b Shows the test NDCG@10 when varying whether hash codes are sampled stochastically or deterministically while training and for evaluation. For example, *Det.Eval + Stoc.Train* corresponds to deterministic sampling of hash codes for evaluation, while sampling stochastically when training the model.

Convergence plots for the remaining datasets are shown in the Appendix, where we observe the same trend.

**Stochastic or deterministic sampling.** We investigate the effect of the sampling strategy for the hash codes (see Eq. 10) during training and evaluation. The sampling can either be deterministic ($\mu^{(j)} = 0.5$) or stochastic ($\mu^{(j)}$ is sampled uniformly at random from $[0, 1]$), and does not have to be the same for training and evaluation. Figure 2b shows the performance for these 4 configurations across all datasets. We see that stochastic training with deterministic evaluation performs the best, while deterministic training and deterministic evaluation perform second best. As expected, stochastic sampling at evaluation performs significantly worse than deterministic sampling, as every item has a very small probability of being sampled such that it has a small Hamming distance to a user, even though it has a low rating (and vice versa for highly rated items).

### 4.6 Runtime Analysis

Self-masking has an additional cost added to the standard Hamming distance, due to the additional `AND` operation between the user and item hash codes (see Eq. 1 and 2). We now investigate the actual runtime cost associated with this modification.

We implement both the Hamming distance and Hamming distance with self-masking efficiently in C on a machine with a 64 bit instruction set. A test environment was made with 64 bit hash codes for 100,000 users and 100,000 items. For each user, the distances were computed to all items using both the Hamming distance and Hamming distance with self-masking. We measure the actual time taken for computing the distances to all items from each user, and report the average over 50 repeated runs. All experiments are run on a single thread[10], with all users and items loaded in RAM. The code was compiled with the highest optimization level, and utilizing all optimization flags applicable to the hardware. We verified the produced assembler code used the efficient *popcnt* instruction.

The mean experiment time was 8.0358s when using the Hamming distance, and 8.3506s when using the Hamming distance with self-masking. Thus, self-masking only adds a runtime overhead of 3.91% compared to using the standard Hamming distance. As in this setup we are only computing the distances, this can be seen as an upper bound of the actual overhead in a complete system, as the remaining operations (e.g., sorting) would be the same with and without self-masking. Thus, this provides a good trade-off compared to the large performance gains it yields. Note that the measured times are for the total $10^{10}$ distance computations, highlighting the scalability of hashing-based methods to datasets of massive scale. For comparison, if the experiment is repeated with computing

---

[10]We used a Intel(R) Xeon(R) CPU E5-2670 v2 @ 2.50GHz.

the dot product of floating point vectors of size 64, then the computation time is 648.5632s, thus close to 80x slower.

## 5 CONCLUSION

We proposed an end-to-end trainable variational hashing-based collaborative filtering method, which optimizes hash codes using a novel modification to the Hamming distance, which we call *self-masking*. The Hamming distance with self-masking first creates a modified item hash code, by applying an AND operation between the user and item hash codes, before computing the Hamming distance. Intuitively, this can be seen as ignoring user-specified bits when computing the Hamming distance, corresponding to applying a binary importance weight to each bit, but without using more storage and only a very marginal runtime overhead. We verified experimentally that our model outperforms state-of-the-art baselines by up to 12% in NDCG at different cutoffs, across 4 widely used datasets. These gains come at a minimal cost in recommendation time (self-masking only increased computation time by less than 4%).

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

# A APPENDIX

## A.1 CONVERGENCE PLOTS

Convergence plots for Yelp, Amazon, and ML-10M are shown in Figure 3. We observe a similar trend to ML-1M in Figure 2a, where the self-masking leads to a notably faster rate of convergence.

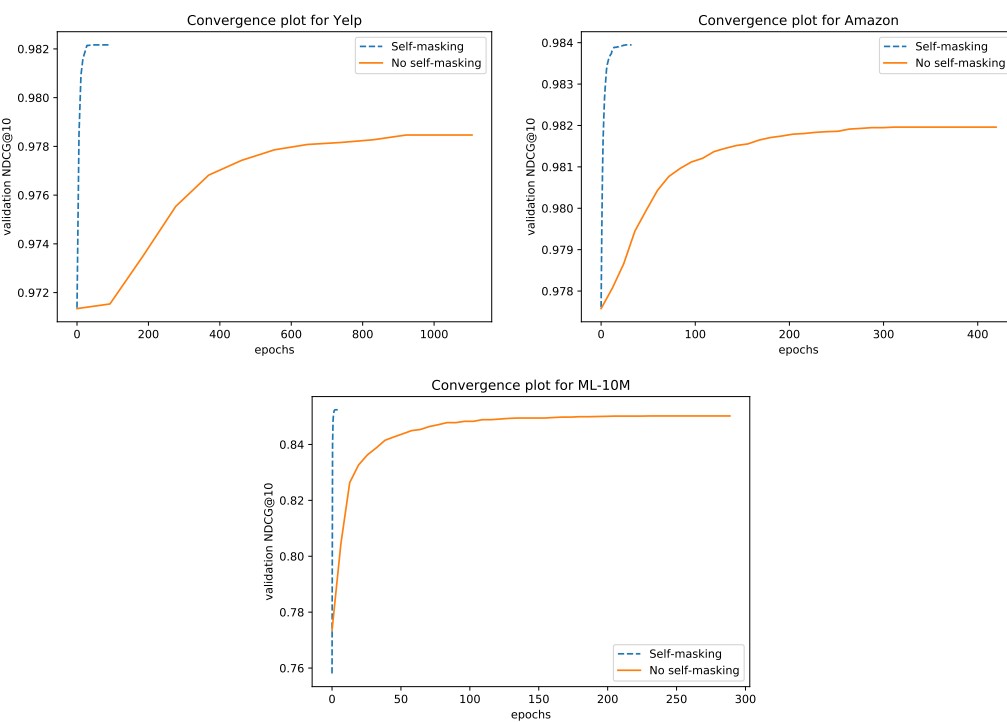

Figure 3: Convergence plot for Yelp, Amazon, and ML-10M.

