# OpenReview forum: "Variational Hashing-based Collaborative Filtering with Self-Masking"
_ICLR.cc/2020/Conference — Reject_

### Official Review · AnonReviewer3 · 2019-10-16
**Official Blind Review #3**

**Rating:** 3

**Review:**

In this paper, the authors study a classical (and well-studied) problem called rating prediction from a new perspective, i.e., learning binary vector representations of the users' and items' latent representations for efficiency. In particular, the authors introduce a personalized self-masking shown in Eq.(2) and in Figure 1 (the 'AND' operation) in order to improve the previous hashing-based collaborative filtering methods without increasing the time cost much.

Empirical studies on four public datasets show the effectiveness of the proposed model, i.e., variational hashing-based collaborative filtering with self-masking (VaHSM-CF).

Some comments/suggestions:

1 The studied problem (i.e., rating prediction) is well studied in the community of recommender systems, and there are many more accurate algorithms than the basic MF algorithm used in the empirical studies. I thus suggest the authors to include more such algorithms though the focus of this paper is for efficiency (but the authors also claim the accuracy of the proposed model).

2 The authors are suggested to give a brief explanation on choosing those baseline methods in the context of other hashing-based CF methods.

3 Some details are missing, e.g., the number of latent dimensions in MF. And some presentation in the parameter setting are not clear, e.g., 'is chosen consistently across all data sets', 'was consistently chosen'. Does 'consistently' means 'exactly the same'?

Minors:
There are some typos: 'a user, u, and', 'of e.g., restaurant and shopping malls', etc.



**Experience Assessment:**

I have read many papers in this area.

**Review Assessment: Checking Correctness Of Derivations And Theory:**

I assessed the sensibility of the derivations and theory.

**Review Assessment: Checking Correctness Of Experiments:**

I carefully checked the experiments.

**Review Assessment: Thoroughness In Paper Reading:**

I read the paper at least twice and used my best judgement in assessing the paper.

---

> ### Author Response · Authors · 2019-11-14
> **Author response**
>
> We thank the reviewer for their insightful review that we have taken into consideration for improving our submission. Below we detail each comment individually.
>
> Q: “1 The studied problem (i.e., rating prediction) is well studied in the community of recommender systems, and there are many more accurate algorithms than the basic MF algorithm used in the empirical studies. I thus suggest the authors to include more such algorithms though the focus of this paper is for efficiency (but the authors also claim the accuracy of the proposed model).”
> A: While this could in principle be included, the focus of our work is on efficient hashing based approaches, and similar to all related work in this and similar domains (described in Section 2, but also that of “semantic hashing” not part of our related work), this is outside the scope of the paper. Thus, we keep an experimental setup similar to related work, where the focus is on the performance among efficient hashing based approaches.
>
> Q: “2 The authors are suggested to give a brief explanation on choosing those baseline methods in the context of other hashing-based CF methods.”
> A: There has been fairly scant research done on hashing based approaches for collaborative filtering with explicit feedback without side information (e.g., content information such as item descriptions or user reviews). As such, DCF and CCCF represent the state-of-the-art in this domain. We have updated parts of section 4.2 to provide a better explanation for our choices.
>     An older two-stage rounding approach named BCCF has been used in related work, however, all current methods have been shown to outperform this approach, which is why we have not included it as part of our experimental setup.
>
> Q: “3 Some details are missing, e.g., the number of latent dimensions in MF. And some presentation in the parameter setting are not clear, e.g., 'is chosen consistently across all data sets', 'was consistently chosen'. Does 'consistently' means 'exactly the same'?”
> A: The latent dimension in MF is the same as the number of bits used, e.g., for hash codes of 32 bits the competing MF would use a latent dimension of 32 as well.
> Yes, “consistently chosen” means exactly the same.
> We have updated the paper to reflect both points.
>
> We hope the above comments clarify the uncertainties, and that the reviewer will reconsider their recommendation.

---

### Official Review · AnonReviewer1 · 2019-10-23
**Official Blind Review #1**

**Rating:** 3

**Review:**

The author proposes a variational encoder (VAE) based approach to perform hashing-based collaborative filtering, which focuses on the weighting problem of each hash-code bit by applying the "self-masking" technique. This proposed technique modifies the encoded information in hash codes and avoids the additional storage requirements and floating point computation, while preserving the efficiency of bit manipulations thanks to hardware-based acceleration. An end-to-end training is achieved by resorting the well-known discrete gradient estimator, straight-through (ST) estimator.

Strength:
The idea of employing the discrete VAE framework to perform hashing-based collaborative filtering is interesting. From the perspective of applications, I think it is somewhat novel.

The experimental results demonstrate the performance superiorities of the self-masking hashing-based collaborative filtering method.



Weakness:

I have concerns over using the VAE to model the ratings of each user and item pairs here. Essentially, you are building a VAE for every rating value R_u, i.  Although the parameters are shared, the VAE’s have their own prior. For generative models like VAE, they need to learn from a lot of data, not just one data point. From the perspective of generating data looking similar to the training data, I don’t think your model have learned anything. Only the reconstruction part is important to your model.

The idea of using discrete VAE for hashing tasks has been explored before, see “NASH: Toward End-to-End Neural Architecture for Generative Semantic Hashing”.  It employed a very similar idea, although it is not used for CF, but for hashing directly. The novelties of the model is limited.

Since ratings are essentially ordinal data, using Gaussian distribution to model the rating data may be not appropriate.

The whole paper, especially the model, is not presented well. The model is not presented in a rigorous way, and some sentences in this paper are difficult to follow.

The runtime analysis is not sufficient. In addition to comparing with the methods using hamming distance, we also want to see the advantages of the hamming based method over the real-value based method on speed acceleration.


Other question:

To realize the self-masking role, the paper proposed to use the function f(z_u, z_i)=g(Hamming_self-mask(z_u, z_i)), Equ. 12, and demonstrate the effectiveness of using this function by experiments. Since the necessity of using self-masking is not very convincing, I doubt whether this self-masking function f(z_u, z_i) is indispensable. Maybe, if some other functions that takes z_u and z_i as input are used, better results may be observed.


**Experience Assessment:**

I have published one or two papers in this area.

**Review Assessment: Checking Correctness Of Derivations And Theory:**

I assessed the sensibility of the derivations and theory.

**Review Assessment: Checking Correctness Of Experiments:**

I assessed the sensibility of the experiments.

**Review Assessment: Thoroughness In Paper Reading:**

I read the paper at least twice and used my best judgement in assessing the paper.

---

> ### Author Response · Authors · 2019-11-14
> **Author response part 2**
>
> Q: “To realize the self-masking role, the paper proposed to use the function f(z_u, z_i)=g(Hamming_self-mask(z_u, z_i)), Equ. 12, and demonstrate the effectiveness of using this function by experiments. Since the necessity of using self-masking is not very convincing, I doubt whether this self-masking function f(z_u, z_i) is indispensable. Maybe, if some other functions that takes z_u and z_i as input are used, better results may be observed. “
> A: The necessity of self-masking is due to the inability to *efficiently* compute a *weighted* Hamming distance, as it is only based on a hardware supported sum (popcount) and a Boolean XOR. Motivated by the hypothesis that not all bit dimensions are equally important for each user, we derive self-masking as a way to do binary weighting. The binary weighting is implemented by masking the item by the user hash code (i.e, using an AND operation), such that the user hash code determines which dimensions are important for the user. After the masking, the Hamming distance is now effectively only computed on the bit dimensions set to 1 in the user hash code (since all items have a value of -1 in dimensions that are -1 in the user hash code).
>     While in principle other functions taking z_u and z_i as input might exist, they would have to be described using only Boolean operations and without using other inputs (otherwise, it would no longer be fast to compute). Thus, this set of possible functions is extremely limited, and we have not been able to find any other meaningful functions than our self-masking.
>     Without self-masking, we are left with our VaH-CF model as seen in Table 2, where the experimental results clearly show the large performance improvement obtained by self-masking.
>
> We hope the above comments clarify the uncertainties, and that the reviewer would reconsider their recommendation.
>
> [1] Chenghao Liu, Tao Lu, Xin Wang, Zhiyong Cheng, Jianling Sun, and Steven C.H. Hoi. 2019. Compositional Coding for Collaborative Filtering. In Proceedings of the 42nd International ACM SIGIR Conference on Research and Development in Information Retrieval (SIGIR'19). ACM, New York, NY, USA, 145-154.

---

> ### Author Response · Authors · 2019-11-14
> **Author response part 1**
>
> We thank the reviewer for their insightful review, which we have taken into consideration for improving our submission. Below, we detail each comment individually.
>
> Q: “I have concerns over using the VAE to model the ratings of each user and item pairs here. Essentially, you are building a VAE for every rating value R_u, i.  Although the parameters are shared, the VAE’s have their own prior. For generative models like VAE, they need to learn from a lot of data, not just one data point. From the perspective of generating data looking similar to the training data, I don’t think your model have learned anything. Only the reconstruction part is important to your model. “
> A: It is correct that only the reconstruction part is important in our case, since we do not need the generative properties of the model for its use-case of collaborative filtering. As described in Section 4.1, and similarly done in related work, we require each user to have at least 10 ratings, and each item to be rated at least 10 times, such that the model does have data to learn the dynamics between users and items. While this amount of data may not be sufficient to utilize the generative properties, we find that it is sufficient for learning hash codes that enable highly efficient item recommendations. We have extended our revised paper with this information in Section 3.4.
>
> Q: “The idea of using discrete VAE for hashing tasks has been explored before, see “NASH: Toward End-to-End Neural Architecture for Generative Semantic Hashing”.  It employed a very similar idea, although it is not used for CF, but for hashing directly. The novelties of the model is limited.”
> A: Methods based on discrete VAE have indeed previously been used in the task of semantic hashing for both text and images, where text/images are transformed to short hash codes similarly to our work. In contrast to these, we are the first to design a discrete VAE for hashing-based collaborative filtering, whereas existing approaches are based on traditional non-neural setups. However, the most important contribution of our work is the introduction of self-masking in the Hamming distance. While this may appear to be minor, we are the first to consider changes to the Hamming distance which we experimentally show to significantly improve performance.
>
> Q: “Since ratings are essentially ordinal data, using Gaussian distribution to model the rating data may be not appropriate.”
> A: We use a Gaussian distribution to model the difference between the observed and estimated ratings. While one option would be to use different variances depending on the rating, this would effectively enforce different weights in the reconstruction. We make the choice of weighting all ratings equally and thus fix the variance to be the same across all ratings. As such, we do not agree that it is inappropriate, and in fact this leads to an MSE objective identical to our baselines (which implicitly make the same Gaussian assumption). We have updated Section 3.4 in our revised paper to reflect this.
>
> Q: “The whole paper, especially the model, is not presented well. The model is not presented in a rigorous way, and some sentences in this paper are difficult to follow.”
> A: We have updated the paper and made small changes to improve the presentation.
>
> Q: “The runtime analysis is not sufficient. In addition to comparing with the methods using hamming distance, we also want to see the advantages of the hamming based method over the real-value based method on speed acceleration.”
> A: We have now added the floating-point case using the dot product (see Section 4.6).

---

### Official Review · AnonReviewer2 · 2019-10-29
**Official Blind Review #2**

**Rating:** 1

**Review:**

In this paper, the authors proposed an end-to-end variational hashing-based collaborative filtering scheme with self-masking to solve the efficiency of large-scale recommender systems. It mainly targets at addressing a problem in existing hashing-based collaborative filtering methods, where each bit is equally weighted in the Hamming distance computing process. To this end, the presented hashing scheme develops a self-masking technique to encode which bits are important to the user. The comparative experiments on several datasets demonstrate the superior performance of the presented hashing method.

The idea is interesting in the sense that an efficient self-masking technique is proposed to generate user-adaptive hash codes, by differentiating the importance of binary bits, for a fast recommendation system. However, the issues with the experimental results make me inclined to reject this paper.

In Table 2, the reported results of MF and its variants are lower than those of DCF. These results are very unreasonable. I have also conducted many similar experiments and the obtained results are not consistent with the results this paper reported.

In addition, in the original paper, CCCF is reported to achieve the superior performance over DCF. However, in this paper, the results are also inconsistent.

To sum up, the reported results in the paper are not convincing to me. I would doubt that there is no much accuracy improvement compared against the state-of-the-art methods.

Finally, as I understand, using variational hashing to maximize the likelihood of all observed items and users will inevitably increase the training time. The presented experiments, however, include no results on the efficiency comparison.

If there are no advantages on the recommendation accuracy and efficiency, the motivations of this designed hashing method are vague.

Based on the above reasons, I tend to reject this paper.

**Experience Assessment:**

I have published in this field for several years.

**Review Assessment: Checking Correctness Of Derivations And Theory:**

I assessed the sensibility of the derivations and theory.

**Review Assessment: Checking Correctness Of Experiments:**

I carefully checked the experiments.

**Review Assessment: Thoroughness In Paper Reading:**

I read the paper at least twice and used my best judgement in assessing the paper.

---

> ### Author Response · Authors · 2019-11-14
> **Author response**
>
> We thank the reviewer for their insightful review, which we have taken into consideration for improving our submission. Below we detail each comment individually.
>
> Q: “In Table 2, the reported results of MF and its variants are lower than those of DCF. These results are very unreasonable. I have also conducted many similar experiments and the obtained results are not consistent with the results this paper reported.”
> A:  We used the code as provided in the CCCF [1] GitHub repository [2]. However, after your comment, we found that a wrong distance measure had been used for MF, and this has now been correctly changed to the dot product. We have updated the MF baseline results in the revised paper, and the performance is now notably higher, as expected (thus reflecting the performance scores observed in previous work). This, however, does not change the performance comparison against the state-of-the-art hashing based approaches, which is the focus of the paper and where our approach still leads to a very substantial improvement.
>
> Q: “In addition, in the original paper, CCCF is reported to achieve the superior performance over DCF. However, in this paper, the results are also inconsistent.”
> A: CCCF learns floating-point weights that are used for the weighted Hamming distance computation. In their original paper, the authors (mistakenly, we believe) do not count these floating-point weights when counting the total number of bits for each hash code, thus leading to an unfair experimental comparison. For example, they use 8 blocks for all hash code lengths, which corresponds to 128-512 (depending on the floating-point precision) additional unaccounted bits compared to the baselines (this is a very significant difference when using hash codes of length 32-64).
>     When we account for the floating-point weights, CCCF obtains similar performance to DCF, which is to be expected, since single-block CCCF has an almost identical formulation as DCF. As described in our Section 4.3, we experiment with block sizes of {8,16,32,64} and count each floating-point as only 16 bits — we try all combinations within the bit budget (32 and 64) and report the best performance (if a single block is chosen, we do not count the floating-point weight). We focus on hash code lengths of 32 and 64 as these correspond to the common machine word lengths, and thus the case where the Hamming distance is fastest to compute (compared to cases where the hash code spans multiple machine words).
>
> Q. “To sum up, the reported results in the paper are not convincing to me. I would doubt that there is no much accuracy improvement compared to the state-of-the-art methods.”
> A: We have revised our paper in response to the reviewer’s comments regarding the two questions above. Our results (see Table 2) show our approach does in fact provide a large improvement compared against the state-of-the-art hashing based methods. We hope our comments and paper changes related to the previous questions above clarify this in a satisfactory manner.
>
> Q: “Finally, as I understand, using variational hashing to maximize the likelihood of all observed items and users will inevitably increase the training time. The presented experiments, however, include no results on the efficiency comparison.”
> A: In Section 4.5 we analyze the convergence rate. Our self-mask method enables the model to converge extremely fast (often 5-20 epochs)  compared to the no self-masking case (up to 1000 epochs), which means it is fast to train.
>
> Q: “If there are no advantages on the recommendation accuracy and efficiency, the motivations of this designed hashing method are vague.”
> A: We hope our changes and explanations above provide evidence of the advantages of our approach with regards to both effectiveness and (training/runtime) efficiency, and that the reviewer will revise their recommendation.
>
> [1] Chenghao Liu, Tao Lu, Xin Wang, Zhiyong Cheng, Jianling Sun, and Steven C.H. Hoi. 2019. Compositional Coding for Collaborative Filtering. In Proceedings of the 42nd International ACM SIGIR Conference on Research and Development in Information Retrieval (SIGIR'19). ACM, New York, NY, USA, 145-154.
> [2] https://github.com/3140102441/CCCF

---

### Decision · Program_Chairs · 2019-12-19

**Decision:**

Reject

**Comment:**

There was a clear consensus amongst reviewers that the paper should not be accepted. This view was not changed by the rebuttal. Thus the paper is rejected.